# The effect of exchanging drawings with peers on the happiness of children with cancer, aged 7–11 years: A clinical trial

Somayeh Palvan[1], Khadijah Zareii[1], Akram Sadat Sadat Hoseini [ORCID][2]*, Hamid Haghani[3]

**1** School of Nursing and Midwifery, Tehran University of Medical Sciences, Tehran, Iran, **2** Department of Pediatric and NICU, School of Nursing and Midwifery, and a member of Research Center of Hadith, Quran and Medical Sciences, Tehran University of Medical Sciences, Tehran, Iran, **3** Department of Biostatistics, Shahid Beheshti University, Tehran, Iran

* ashoseini@tums.ac.ir

**Data Availability Statement:** All relevant data are within the paper and its Supporting Information files.

**Funding:** This research did not receive any funding for publication but took grant NO 9311700002 of

## Abstract

### Introduction

Improving the mental status of children with cancer is part of nurses' duties in planning nursing care and is achieved in different ways. This study attempts to combine drawing with peer interaction to improve the mental status of sick children and investigate the effect of exchanging drawings with peers on these children's happiness.

### Method

This clinical trial was conducted on a sample of 66 children with cancer, ages 7–11 years, who were randomly assigned to two groups. The intervention was carried out over five weeks by the exchange of drawings between healthy children at school and children with cancer. Both groups of children drew one drawing each week. The children's happiness was measured by the researcher before and after the intervention (i.e., week five) using a questionnaire.

### Results

The results showed no significant differences between the two groups in terms of happiness after the intervention. The happiness score was 3.15 ±0.34 in the control group and 3.02 ±0.3 in the intervention group before the intervention; afterwards, this score was 3.022± 0.22 among the controls and 3.11± 0.25 among the patients. The comparison of the two groups using the t-test showed P = 0.075 before the intervention and P = 0.11 after the intervention.

### Conclusion

Given the results obtained, future studies are recommended to administer lengthier interventions and enable the physical presence of healthy peers along sick children or to use the exchange of drawings with peers in combination with other psychological interventions so as to promote happiness in children with cancer.

Tehran University of Medical Sciences for research process.

**Competing interests:** No authors have competing interests.

## Introduction

Studies have shown that developing cancer at a young age puts children at a higher risk of psychological disorders or other problems concerning their social compatibility with peers [1, 2]. Happiness is a psychological status that is reduced dramatically in children with cancer as a result of their frequent hospitalizations and their difficult, painful and prolonged treatment periods [3–5]. In sick children, happiness has the ability to reduce emotional and behavioral problems and control depression [6–8]. In general, happiness leads to a positive attitude toward life, mental health, emotional balance and a better functioning of the immune system in dealing with stresses [3, 9, 10]. Positive psychology is currently a promising approach for improving the quality of life in patients [3, 11, 12]. Happiness often diminishes dramatically in children with cancer due to the nature of the disease and its treatment, and since happiness has a major role in children's growth and development, it is a vital part of nursing care [7, 8, 12–17]. Arts can be used as one of the ways to improve happiness [18].

Drawing can affect children's feelings and enable the expression and release of emotions. Some studies have emphasized the effectiveness of drawing in improving children's psychological status [18–21]. In drawing sessions, children express their negative feelings through drawing and regulate their emotions by projecting their feelings [4, 22, 23]. A child's drawing is a message; that is, children express their emotional and affective world by means of drawing, which is a language for the expression of feelings [4, 22]. Drawing is therefore a way of expressing feelings for children that affects their mental status, and interaction with peers can further improve children's mental status. Studies have shown that peers learn, help and complement one another [10, 18, 24]. A peer is someone who has similarities with the patient in a number of features, such as age, gender, occupation, socioeconomic status and health status [24, 25]. Several studies have demonstrated the effect of peer groups as a source of philanthropic, emotional and educational support and have recommended this complementary approach in conjunction with other care-promoting strategies [10, 26]. A meeting between peers is a source of relief and reassurance for patients that can encourage them to learn coping strategies to overcome their disease [24, 25]. Given the peer influence during childhood, this potential can probably help nurses further improve their care measures although no research has yet been conducted on the effect of healthy peers on sick children [27, 28]. Further studies are therefore required on how the provision of psychological and social support to patients, such as the support of peers, can improve the conditions of children with cancer. This study therefore combines two care measures, namely drawing and interaction with peers, and assesses their cumulative outcome.

Drawing is a means of expressing feelings in children that can be used to help with their psychological problems, as well [22]. School-aged children have a particular need for interacting with their peers, which can be disrupted as a result of illness and hospitalization, treatment and prolonged recovery [16, 24, 25]. To ensure the proper growth and development of children, nurses are responsible for facilitating children's interaction with peers in their care planning [29]. Designing care procedures that simultaneously account for both of these interventions requires creativity and innovativeness. This study devised a care plan consisting of the exchange of drawings drawn by children with cancer and healthy children so as to convey happiness to children with cancer. This care plan was chosen to facilitate the children's expression of feelings through drawing and to enable the sick children's interaction with their peers through the messages conveyed by the drawings. The effectiveness of this case intervention, which has been less addressed in studies on children with cancer, is assessed based on positivist psychology and the measurement of the children's happiness and the subsequent boost in their spirits. This study thus examines the effect of the exchange of drawings with

peers on the happiness of children with cancer and assesses the effect of combining the drawing method and interaction with peers on the happiness of these children.

## Materials and methods

For this clinical trial, 66 children with cancer aged 7–11 years were selected (based on sample size formula for comparison of two means). At a significance level of 0.05, test power of 80%, and also assuming that the effect of peer drawing on happiness of school-aged children with cancer is 1.8 (10% of the maximum score of the instrument), sample size was determined 30 children per group. Based on the range of happiness score from 0 to 87, standard deviation was estimated at 14. The sample size was increased by 10% to compensate possible withdraws. Thus, the final sample size per group was determined n = 33.

The children with cancer were selected through convenience sampling and were then randomly assigned to the control and intervention groups by block randomization, and two modes, A and B, were considered to indicate the control (A) and intervention (B) groups. The following six permutations were obtained for these two modes, including (1) BAAB; (2) ABAB; (3) AABB; (4) ABBA; (5) BBAA; and (6) BABA. Next, 17 blocks of four were randomly sampled by throwing a six-sided dice (17 modes for 66 children). Blocks were prepared by a statistician and sampling was carried out by the research team according to predetermined sequence with no bias. The samples were selected using convenience sampling method from hospitalized children in the oncology ward of Children's Medical Center, such that eligible children were assigned to control (A) and intervention (B) groups. Sampling was carried out almost every day with inclusion of eligible children, who were allocated to a group on the day of admission.

The children in the peer group were selected through census sampling; that is, any child of the intended age range studying at the noted school and willing to take part in the study who had their parents' informed consent was included. The Ministry of Education granted permission to conduct the study.

The study inclusion criteria for the intervention and control groups consisted of being aged 7–11 years, having a definitive diagnosis of ALL, having medical records available at the hospital, the elapse of at least three months from the diagnosis of cancer, having no physical or mental disability that impeded drawing and being hospitalized and receiving treatment and medications at the oncology ward at least every two weeks.

The exclusion criteria for intervention and control groups consisted of absence from more than two sessions of drawing.

The data collection tools included a demographic questionnaire and the Children's Happiness Scale.

The demographic questionnaire inquired about the children's age, gender, birth order, time of diagnosis of cancer, underlying diseases and interest in and habits concerning drawing.

The Children's Happiness Scale was developed in 2014 by Dr. Rodger Morgan and consists of 20 graded items scored by children; that is, children check any items they agree with in relation to themselves, and the scores are then summed up. The sum of the scores obtained is then divided by the number of checked items to find the child's happiness score. The highest score (i.e. happiest) is 4.25 and the lowest is 1.68.

The original tool was in English, and the tool developer gave the researchers permission to translate and use it. First, the tool was translated into Persian, and its translation was approved by three people familiar with English and Persian. To examine the validity of the scale, the face and content validity method was used. For this purpose, after translation, the scale was distributed among ten psychology and nursing professors to review and confirm its items. The

experts' comments were then collected and analyzed, and necessary modifications were made to the items in terms of wording. The scale reliability was assessed using test-retest method, such that it was distributed among ten sick children (they were excluded from the study) selected by convenience sampling, who completed the items at baseline and then again a week later. The correlation between the two measurements was 7.8, which indicates an acceptable reliability [7–9, 30]. The present study was approved by the Ethics Committee of School of Nursing and Midwifery and Rehabilitation of Tehran University of Medical Sciences on 11.9.2017. Validation of the tool began on 20.11.2017, and pilot sampling for assessing reliability began on 4.4.2018. The initial pilot study lasted a month, and sampling and intervention began on 7.9.2018. Since schools were closed on the anticipated date in RCT, and children's peers first became available in school on 7.9.2018, the intervention and sampling lasted three months. Hence, the time lapse of sampling registration at RCT was due to the initial pilot study, and after obtaining the code of ethics and before final confirmation of RCT, sampling began for validity and reliability assessment of the tool and pilot study. Sampling began after the final confirmation of RCT. (Fig 1)

## Intervention

**Healthy children's drawing.** The researcher visited the selected primary school holding an official permission from the local office of the Ministry of Education, explained the study objectives to the school authorities and then obtained the principal's permission for the students to draw drawings for a group of children with cancer. A meeting was then arranged to brief the children's teachers and parents on the study objectives and methods, and to obtain their informed consent. Once they signed informed consent forms and with prior arrangement with each teacher, 33 students were selected from the second to fourth grades for drawing period. After giving a simple explanation to them about the study objectives and methods in the presence of their teacher, they were asked to draw a drawing with any subject they desired for hospitalized children with cancer once a week for a total of five weeks during their art class. The researcher provided the children with all the necessary tools for drawing including A4 paper, black pencils, colored pencils, colored markers, and colored crayons. Additional pieces of A4 paper were given to them if they wanted to draw more than one drawing. The children finished their drawings over 20–25 minutes on average, and their drawings were collected at the end of each session. This protocol continued once a week for a total of five weeks, and the drawings were taken to the hospital by the researcher every week.

**Children with cancer drawing.** The researcher visited the ward with a letter of introduction from and prior arrangement with the hospital director and the authorities of the nursing office and oncology ward of the Children's Medical Center. Convenience sampling was initially carried out for both the control and intervention groups (n = 33 per group) based on the study inclusion criteria. After briefing them on the study objectives and methods, the parents gave their informed written consent, and the children gave their informed verbal consent. Data were collected before the intervention. The demographic questionnaire was completed by one of each child's parents, and the Children's Happiness Scale was filled out by the children. The children's happiness was also assessed a week after the end of the intervention in both the intervention and control groups.

**Drawing sessions in the hospital.** The drawing intervention was carried out over five consecutive weeks with one-week intervals. The researcher took 10–15 min to explain to the intervention group that they would receive drawings from their peers at a school once a week for a total five weeks and that, in response, they would draw a drawing on any subject of their choice to give to the researcher every week, too. The duration and conditions of the drawing

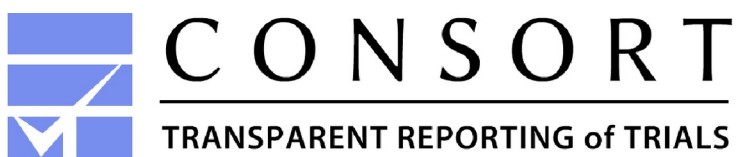

## CONSORT 2010 Flow Diagram

**Enrollment**

Assessed for eligibility (n=66 )

Excluded (n=0  )
♦ Not meeting inclusion criteria (n=0  )
♦ Declined to participate (n=0  )
♦ Other reasons (n=0  )

Randomized (n= 33)

**Allocation**

Allocated to intervention (n=33 )
♦ Received allocated intervention (n= 0 )
♦ Did not receive allocated intervention (give reasons) (n=0  )

Allocated to intervention (n=33 )
♦ Received allocated intervention (n= 0 )
♦ Did not receive allocated intervention (give reasons) (n= 0 )

**Follow-Up**

Lost to follow-up (give reasons) (n=0 )

Discontinued intervention (give reasons) (n=0 )

Lost to follow-up (give reasons) (n= 0)

Discontinued intervention (give reasons) (n= 0 )

**Analysis**

Analysed (n= 33)
♦ Excluded from analysis (give reasons) (n=0 )

Analysed (n= 33)
♦ Excluded from analysis (give reasons) (n=0 )

**Fig 1. Participant flow chart.**

sessions were the same in the hospital as in the school period. The only difference was the setting, as the sick children drew in the hospital's play room. The control group drew under the same conditions as the intervention group, but they received no drawing from their peers and were not led to assume that they were drawing for their peers. The researcher took the intervention group's drawings to the school every week and gave them to the peer group during their art class, and then collected the school children's newest drawings and took them to the hospital for the intervention group children. This process continued for five weeks. The mothers' phone numbers were collected to coordinate the next sessions. On a few occasions when the interval between the sessions was more than one week, the mothers were asked to show their children the drawings that their peers had drawn for them. In these cases, peer drawings were emailed to sick children and the researcher asked them to draw a drawing for their healthy peers, as per the plan, and to send the new drawing to the researcher or bring it to the next session.

To facilitate drawing in the hospital, the drawing sessions were held in the play room in the presence of the researcher. The drawing intervention sessions were held after the child had received their medications and been visited by the doctors to avoid disturbing the child during the process of drawing. The mothers and relatives accompanied the child in the play room when the child was drawing but they had been advised not to interfere with, guide, or comment on the child's drawing. The intervention and its assessment were carried out by the researchers, including a psychiatric nurse, an expert nurse in use of play and an MSc nursing student.

The participants were blinded to their group allocation, and were only told that they were to draw for their peers. To avoid information contamination, the two groups' children entered the play room separately. Researchers assigned children to two groups by block randomization, but were aware of type of intervention for each group, and efforts were made this would not affect children's intervention and type of drawing through necessary training. All data were entered into the software by a blinded person, and test data were analyzed by a blinded statistician.

## Statistical analysis

Data were analyzed in SPSS-16 (SPSS Inc. Released 2007. SPSS for Windows, Version 16.0. Chicago, SPSS Inc) using the independent t-test and paired t-test for comparison of the happiness scores between the two groups (2-tailed), the Chi-square test for comparison of the demographics, Fisher's exact test for interest in drawing and paired t-test for comparison of the happiness scores in each group. Kolmogorov-Smirnov test was used to assess normal distribution of the data. Significance level was assumed 95% for all tests.

## Ethical considerations

For the purpose of collecting data, the researcher first obtained permission from the ethics committee of Tehran University of Medical Sciences (IR.TUMS.FNM.REC.1396,3441) and registered the study (IRCT20150928024239N4).

## Results

The Kolmogorov-Smirnov test showed the normal distribution of the data. The two groups were also compared using parametric tests. No significant differences were observed between the two groups in terms of demographic details and they matched in this regard (S1 Table).

The results showed no significant differences in the children's happiness before and after drawing in the intervention and control groups. There were also no significant differences in

the mean happiness scores between the intervention and control groups before and after the intervention (S2 Table).

## Discussion

The results showed no significant differences between the control and intervention groups in terms of happiness before the intervention, and the level of happiness was not high in either group. The happiness score of the children in the control and intervention groups was indicative of the level of psychological damage to children with cancer, which reveals the urgent need of this group for help to boost their spirits and happiness if they are to make progress in their process of recovery from the disease [31].

Developing cancer in childhood can put the child's mental health at risk. Children experience some painful side-effects as a result of cancer treatment, and because of illness, they can no longer attend their school or participate in the gatherings of their family and friends. Their life therefore undergoes enormous changes [32]. These children are also predisposed to infection and are continually hospitalized, which leads to their isolation from the family and society [21, 32]. These children are therefore denied of the particular sense of happiness that is characteristic of childhood; however, happiness and vitality make treatment more successful and increase life expectancy [33]. The results obtained by Sanjari et al. showed that, following the diagnosis of cancer, adolescents experience higher levels of anxiety and depression and lower levels of vitality and mental health [34].

There were no significant differences between the control and intervention groups in the level of happiness after the drawing intervention. The intervention consisting of the exchange of drawings with peers thus failed to exert any effect on the happiness of sick children. In the researcher's view, the ineffectiveness of the intervention and the homogeneity of the happiness level between the two groups after the intervention could be due to the physical absence of the peer group. The results obtained in peer-based studies confirm this claim. When such interventions were conducted in the presence of both the peer and intervention groups, a significant difference was reported between the intervention and control groups [24, 28, 29].

The lack of a significant difference in the happiness of sick children after the drawing intervention can motivate future studies to use other interventions in combination with drawing, such as music therapy, relaxation exercises and play therapy. These combination interventions may increase happiness in children with cancer. The concurrent use of a number of interventions, such as drawing, music, puppet shows and games appears to have greater effects on the happiness of hospitalized children.

Given that more studies have confirmed the effectiveness of drawing in reducing anxiety and depression in sick children [19, 35], the non-significant effect obtained in the present study for drawing in relation to children's happiness reveals a number of issues. First, the use of drawing might be promoted in healthcare settings, and the research group recommends at least 12 sessions, lasting 45 to 60 minutes each. In a study conducted by Lee, 14 weekly sessions of drawing therapy had positive effects on changing the behavior and emotions of children with behavioral problems [36]. Although the process and goals of drawing therapy and drawing are not similar, this research showed benefit of the process of drawing for reducing behavior problems. Moreover, meetings held between peers act as a source of relief and reassurance for patients and can encourage them to learn coping strategies to overcome the disease and have a greater chance in life.

In the present study, the sick children and their peers had no face-to-face contact with one another and only communicated through drawings, which appears to have not yielded a significant effect. Another issue that may explain why the exchange of drawings with peers had no

significant effect on the happiness of the sick children is the use of drawing alone without any additional interventions.

As demonstrated in studies by Gariepy et al. and Barrera et al., combining drawing with play leads to significant positive effects [37, 38]. It can probably be concluded that the happiness of sick children cannot be promoted by the mere use of drawing and other methods should also be used in conjunction with drawing for the intervention to prove effective. Future studies are recommended to be conducted with a larger number of intervention sessions along with a longer follow-up instead of mere drawing. We recommend other methods in conjunction with drawing. Also, face-to-face interactions between sick children and their peers should be facilitated in order to increase the effect of the intervention and obtain the intended results.

Acting as intermediaries between healthy and hospitalized children, nurses can change the rules and initiate the presence of peers in hospital settings so as to turn these interactions into effective therapeutic contacts, and healthy children can also gain valuable experience about having a fruitful presence in the society through their interaction with patients.

## Study limitations

The concurrent hospitalization of the intervention and control group children in the ward increased the possibility of being influenced by each other, which could not be avoided, as there were no similar settings available to separate the two groups. The novelty of the subject for the school children made interaction difficult in the first few days because they had no idea about drawing for sick children, but this issue was resolved over time. It would have been better if the drawings had been interpreted in terms of how they promoted happiness, since the interpretation of drawings might show changes in the level of happiness. The physical presence of peers could have increased the effectiveness of the intervention, but this measure was not possible due to the hospital's visiting regulations. If, along with the positivist psychological factors, psychological problems such as anxiety and depression had also been measured, then the effectiveness of this intervention in reducing the psychological problems caused by disease could also have been assessed. Moreover, it would have been better if the healthy children's happiness had also been measured before and after the drawing intervention.

## Conclusion

Although the present study showed no significant differences between the two groups, this innovative technique can be developed through further studies after the shortfalls of this study are resolved. It can then be used as a way of interaction for these children and their healthy peers and to teach healthy individuals about proper interaction with sick children as part of their social responsibility.

## Supporting information

**S1 Checklist. CONSORT checklist.**
(DOC)

**S1 Table. A comparison of the demographics variable.**
(DOCX)

**S2 Table. A comparison of the happiness scores.**
(DOCX)

**S1 File.**
(DOC)

**S2 File.**
(DOCX)

## Author Contributions

**Data curation:** Somayeh Palvan, Hamid Haghani.

**Formal analysis:** Somayeh Palvan, Hamid Haghani.

**Funding acquisition:** Somayeh Palvan.

**Methodology:** Khadijah Zareii, Akram Sadat Sadat Hoseini.

**Project administration:** Somayeh Palvan, Akram Sadat Sadat Hoseini.

**Resources:** Akram Sadat Sadat Hoseini.

**Supervision:** Khadijah Zareii, Akram Sadat Sadat Hoseini.

**Validation:** Akram Sadat Sadat Hoseini.

**Writing – original draft:** Akram Sadat Sadat Hoseini.

**Writing – review & editing:** Akram Sadat Sadat Hoseini.

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
