## [Decision Letter · Decision Letter 0]

27 Oct 2020

PONE-D-20-10349

The Effect of Exchanging Paintings with Peers on the Happiness of Children with Cancer, Ages 7-11 Years: Clinical trial

PLOS ONE

Dear Dr. Sadat Hoseini,

Thank you for submitting your manuscript to PLOS ONE. After careful consideration, we feel that it has merit but does not fully meet PLOS ONE’s publication criteria as it currently stands. Therefore, we invite you to submit a revised version of the manuscript that addresses the points raised during the review process.

Two of the reviewers highlighted fundamental problems with both methodology and resutt description. Consider if by convincingly defending, also statistically, your study design and providing the missing data and information listed in the reviewers' comments you can satisfy the reviewers, otherwise consider retracting the submission and re-submit after the necessary changes have been implemented.

We look forward to receiving your revised manuscript.

Kind regards,

Andrea Martinuzzi

Academic Editor

PLOS ONE

Journal Requirements:

2. Please include additional information regarding the survey or questionnaire used in the study and ensure that you have provided sufficient details that others could replicate the analyses. For instance, if you developed a questionnaire as part of this study and it is not under a copyright more restrictive than CC-BY, please include a copy, in both the original language and English, as Supporting Information.  If the original language is written in non-Latin characters, for example Amharic, Chinese, or Korean, please use a file format that ensures these characters are visible. If the questionnaire can not be published CC-BY, please include a reference.

3. Thank you for submitting your clinical trial to PLOS ONE and for providing the name of the registry and the registration number. The information in the registry entry suggests that your trial was registered after patient recruitment began. PLOS ONE strongly encourages authors to register all trials before recruiting the first participant in a study.

1) your reasons for your delay in registering this study (after enrolment of participants started);

2) confirmation that all related trials are registered by stating: “The authors confirm that all ongoing and related trials for this drug/intervention are registered”.

Please also ensure you report the date at which the ethics committee approved the study as well as the complete date range for patient recruitment and follow-up in the Methods section of your manuscript.

"This study was part of a M.S. thesis supported by Tehran University of

Medical Sciences grant NO 9311700002 and IRCT registration number:

IRCT20150928024239N4"

"no"

Reviewers' comments:

Reviewer's Responses to Questions

**Comments to the Author**

1. Is the manuscript technically sound, and do the data support the conclusions?

Reviewer #1: Partly

Reviewer #2: Yes

Reviewer #3: Partly

2. Has the statistical analysis been performed appropriately and rigorously? 

Reviewer #1: No

Reviewer #2: Yes

Reviewer #3: Yes

3. Have the authors made all data underlying the findings in their manuscript fully available?

Reviewer #1: Yes

Reviewer #2: Yes

Reviewer #3: No

4. Is the manuscript presented in an intelligible fashion and written in standard English?

Reviewer #1: Yes

Reviewer #2: Yes

Reviewer #3: Yes

5. Review Comments to the Author

Reviewer #1: The manuscript entitled ‘The Effect of Exchanging Paintings with Peers on the Happiness of Children with Cancer, Ages 7-11 Years: Clinical trial’ with the aim to examine the effect of the exchange of paintings with peers on the happiness of children with cancer and assesses the effect of combining the painting method and interaction with peers on the happiness of these children.

Comments

Materials and Methods

Sample size calculation

Page 6, information on the sample size calculation to be provided i.e. alpha, beta, 1 or 2 tailed test, outcome measure, attrition rate consideration etc

Page 6, information on who prepared the randomization block (sequence generation/allocation, concealment) and assigned to the groups to be clearly stated.

Page 6, the word control group or peer group to be used systematically. The write-up on the subjects selection to be placed on top first before the randomization process.

Page 7, the sentence 'The exclude their data from analysis for both groups consisted of absence from more than two sessions of painting or death of the child.' to be revised.

Page 7, the language version of the Children's Happiness Scale used in the study to be stated. If it is Iranian version the validation information to be provided/cited.

Page 7, information on blinding (providers, subjects, person assessing the outcome) to be stated.

Page 7, is the lowest score for Children's Happiness Scale 1.67 or 1.68?

Page 7, for the reliability test of the scale, the exact name of the correlation test to be stated. The range of the correlation values to be provided.

Page 10, the sentence ‘In this cases were sent peer’s painting through email’ & ‘Researcher ask their to draw a painting’ to be revised.

Statistical analysis

Page 10, a sub-title for the statistical analysis to be provided.

Page 10, proper citation of SPSS and publisher name to be provided. The use of the independent T-test, the Chi-square test, Fisher's exact test and the paired T-test and its purpose to be clearly stated.

The acceptance level of significance to be stated.

Page 10, study limitation to be placed in the discussion section.

Results

Page 12, Kolmogorov-Smirnov test to be stated in the statistical analysis section.

Page 12, the results with no significant differences in the children's happiness before and after painting in the control group to be stated in the text.

Page 12, the write-up of the results section is too short.

Page 19 N for each group to be stated for Table 1 and 2.

Page 19 Table 1, the alignment of the table and words to be improved. Decimal point for p value to be provided and standardized. Chi-square test, Fisher’s Exact Test to be denoted in table/table footnote. Symbol % for individual figure to be omitted since it has been highlighted in the group name. Word P value to be stated in the last column.

Page 19 Table 2, t-statistic value, df to be placed in another column/row from the p value. Alignment of the table and words to be improved.

It would be good to display/describe the number of subjects who were not happy/happy before intervention and after intervention other than describing the mean score of the scale.

Page 12 & 19, the focus to be more on within group comparison rather than comparison between the groups at each time period since the two groups of subjects are different. Mean difference (pre-post) from each group could be used for between group comparison. Effect size i.e. Cohen’s d and 95% confidence interval to be provided.

Page 19, more baseline characteristics of the study subjects to be provided such as types/ severity of cancer (stages), type of medications, severity of pain, surroundings/environment etc

Some references did not conform to the journal format. Bracket symbol ( ) for reference in the text to be replaced with [ ].

For the CONSORT Flow diagram, group name, assessment period, outcome measure(s) to be included.

The write-up of the manuscript can be further improved in terms of grammar and presentations.

Reviewer #2: Very creative, interesting, and well done study. Congratulations! Sorry that you didn't get the results you were hoping for, but you have done an excellent job in analyzing the limitations and how such a study could be done differently to possibly achieve more positive results.

A few comments:

1. The term "painting" is used throughout. However, the list of materials used does not include paint (or paint brushes), which defines painting for many people. From the materials used, I believe the term "drawing" is more accurate. Thus, the title would be "The Effect of Exchanging Drawings...." and other replacements for "painting" and "paintings" made throughout the manuscript.

2. Page 10, Line 10, "an expert in art-therapy." In many countries the term art therapy means something very specific: Art therapy is a regulated mental health profession that requires a master's or doctoral level education in Art Therapy, clinical hours, supervision. and licensure. There is a credentialing and certification process. As I did not see art therapy credentials after any of the authors' names, I'm assuming that to say art therapy could be misleading to an international audience. The phrase "an expert nurse in the therapeutic use of the arts" would be more accurate and help to avoid confusion and criticism from the art therapy community.

3. A question: the children were instructed that they could draw whatever they wanted. Were the school children told that the purpose of the drawings were to try to make children in the hospital happy? Please clarify.

4. Gender bias: avoid the use of "he" in the general sense, instead use "he/she" or "he and she" or rewrite the sentence in the plural and using "they."

I encourage you to continue to explore and expand on this exciting peer to peer intervention.

Reviewer #3: The purpose of this study is to examine the effect of exchanging paintings with peers on the happiness of children ages 7-11 years using a clinical trial. It sounds like an innovative technique to improve the happiness of children; however, the study design could not provide convincing results for health professionals to further investigate if this technique would really work. Therefore, better study design is needed to improve children’s psychological health.

6. PLOS authors have the option to publish the peer review history of their article (what does this mean?). If published, this will include your full peer review and any attached files.

Reviewer #1: No

Reviewer #2: No

Reviewer #3: No

---

## [Author Response · Author response to Decision Letter 0]

17 Dec 2020

Response to reviewer

Dear Editor,

We are grateful for your meticulous review of this manuscript, and your suggestions, which have been very helpful in improving its quality. We also thank the reviewers for their careful reading and valuable suggestions. All the comments we received on this study have been addressed and replied separately. We hope that these changes to the manuscript will facilitate the decision to publish this study in your journal.

reviewer comments Correction and responses Page and place 

editor . Please ensure that your manuscript meets PLOS ONE's style requirements, including those for file naming. corrected All file

 Please include additional information regarding the survey or questionnaire used in the study and ensure that you have provided sufficient details that others could replicate the analyses. For instance, if you developed a questionnaire as part of this study and it is not under a copyright more restrictive than CC-BY, please include a copy, in both the original language and English, as Supporting Information. If the original language is written in non-Latin characters, for example Amharic, Chinese, or Korean, please use a file format that ensures these characters are visible. If the questionnaire can not be published CC-BY, please include a reference They are attached and explained in the text 

 Thank you for submitting your clinical trial to PLOS ONE and for providing the name of the registry and the registration number. The information in the registry entry suggests that your trial was registered after patient recruitment began. PLOS ONE strongly encourages authors to register all trials before recruiting the first participant in a study.

1) your reasons for your delay in registering this study (after enrolment of participants started);

2) confirmation that all related trials are registered by stating: “The authors confirm that all ongoing and related trials for this drug/intervention are registered”.

Please also ensure you report the date at which the ethics committee approved the study as well as the complete date range for patient recruitment and follow-up in the Methods section of your manuscript.

 Corrected and added in methods Page9 

 Thank you for stating the following in the Acknowledgments Section of your manuscript:

"This study was part of a M.S. thesis supported by Tehran University of

Medical Sciences grant NO 9311700002 and IRCT registration number:

IRCT20150928024239N4" Thanks 

Added in cover letter about funding 

"no"

 Your ethics statement should only appear in the Methods section of your manuscript. If your ethics statement is written in any section besides the Methods, please move it to the Methods section and delete it from any other section. Please ensure that your ethics statement is included in your manuscript, as the ethics statement entered into the online submission form will not be published alongside your manuscript.

 Corrected Page 11

 Please include captions for your Supporting Information files at the end of your manuscript, and update any in-text citations to match accordingly Corrected 

Reviewer 1 Page 6, information on the sample size calculation to be provided i.e. alpha, beta, 1 or 2 tailed test, outcome measure, attrition rate consideration etc

 Corrected Page7

 Page 6, information on who prepared the randomization block (sequence generation/allocation, concealment) and assigned to the groups to be clearly stated.

 Corrected Page7 

 Page 6, the word control group or peer group to be used systematically. The write-up on the subjects selection to be placed on top first before the randomization process. Corrected Page 7 &8 

 Page 7, the sentence 'The exclude their data from analysis for both groups consisted of absence from more than two sessions of painting or death of the child.' to be revised.

 Corrected Page 8

 Page 7, the language version of the Children's Happiness Scale used in the study to be stated. If it is Iranian version the validation information to be provided/cited.

 Corrected Page8 &9 and supplement in the article the original version of tool

 Page 7, information on blinding (providers, subjects, person assessing the outcome) to be stated.

 Corrected Page12

 Page 7, is the lowest score for Children's Happiness Scale 1.67 or 1.68?

 Corrected Page8 

 Page 7, for the reliability test of the scale, the exact name of the correlation test to be stated. The range of the correlation values to be provided.

 Corrected Page 9

 Page 10, the sentence ‘In this cases were sent peer’s painting through email’ & ‘Researcher ask their to draw a painting’ to be revised.

 Corrected Page12

 Page 10, a sub-title for the statistical analysis to be provided.

 Corrected Page 12

 Page 10, proper citation of SPSS and publisher name to be provided. The use of the independent T-test, the Chi-square test, Fisher's exact test and the paired T-test and its purpose to be clearly stated.

 Corrected Page 12

 The acceptance level of significance to be stated. Corrected Page 12

 Page 10, study limitation to be placed in the discussion section.

 Corrected Page 17

 Page 12, Kolmogorov-Smirnov test to be stated in the statistical analysis section.

 Corrected Page 12 

 Page 12, the results with no significant differences in the children's happiness before and after painting in the control group to be stated in the text.

 Corrected Page 13

 Page 12, the write-up of the results section is too short.

 Yes of course but I explain all criteria in tables 

 Page 19 N for each group to be stated for Table 1 and 2. Corrected In the file of table was corrected

 Page 19 Table 1, the alignment of the table and words to be improved. Decimal point for p value to be provided and standardized. Chi-square test, Fisher’s Exact Test to be denoted in table/table footnote. Symbol % for individual figure to be omitted since it has been highlighted in the group name. Word P value to be stated in the last column.

 Corrected In the file of table was corrected

 Page 19 Table 2, t-statistic value, df to be placed in another column/row from the p value. Alignment of the table and words to be improved.

 corrected In the file of table was corrected

 It would be good to display/describe the number of subjects who were not happy/happy before intervention and after intervention other than describing the mean score of the scale.

 The scale did not have a cutoff point to determine happy/unhappy children and only reported the mean level of happiness. Therefore, the research team could not divide the children into happy and unhappy groups.

 Page 12 & 19, the focus to be more on within group comparison rather than comparison between the groups at each time period since the two groups of subjects are different. Mean difference (pre-post) from each group could be used for between group comparison. Effect size i.e. Cohen’s d and 95% confidence interval to be provided.

 corrected Do in the file of table 2

 Page 19, more baseline characteristics of the study subjects to be provided such as types/ severity of cancer (stages), type of medications, severity of pain, surroundings/environment etc

 We collected and recorded the data that we found in the patients’ files such as the type of cancer (they were all ALL) and that all underwent chemotherapy, but no other information was recorded in their files and unfortunately, we could not report other data. Table 1 in the file of table

 Some references did not conform to the journal format. Bracket symbol ( ) for reference in the text to be replaced with [ ].

 Corrected 

 For the CONSORT Flow diagram, group name, assessment period, outcome measure(s) to be included. Corrected In the file of consort was attached 

 The write-up of the manuscript can be further improved in terms of grammar and presentations Did it In the all section of manuscript

Reviewer 2 

 Very creative, interesting, and well done study. Congratulations! Sorry that you didn't get the results you were hoping for, but you have done an excellent job in analyzing the limitations and how such a study could be done differently to possibly achieve more positive results.

 Thanks for attention I hope in future research we reduce limitation and get positive results 

 . The term "painting" is used throughout. However, the list of materials used does not include paint (or paint brushes), which defines painting for many people. From the materials used, I believe the term "drawing" is more accurate. Thus, the title would be "The Effect of Exchanging Drawings...." and other replacements for "painting" and "paintings" made throughout the manuscript.

 Corrected In the manuscript

 2. Page 10, Line 10, "an expert in art-therapy." In many countries the term art therapy means something very specific: Art therapy is a regulated mental health profession that requires a master's or doctoral level education in Art Therapy, clinical hours, supervision. and licensure. There is a credentialing and certification process. As I did not see art therapy credentials after any of the authors' names, I'm assuming that to say art therapy could be misleading to an international audience. The phrase "an expert nurse in the therapeutic use of the arts" would be more accurate and help to avoid confusion and criticism from the art therapy community.

 Thanks for your valuable points. All the mentioned points are correct and the manuscript was modified accordingly. Page 12

 A question: the children were instructed that they could draw whatever they wanted. Were the school children told that the purpose of the drawings were to try to make children in the hospital happy? Please clarify. No, school children were only told to draw some pictures for children with cancer. The study objectives were explained to children and their parents when we were going to obtain informed consent. 

 Gender bias: avoid the use of "he" in the general sense, instead use "he/she" or "he and she" or rewrite the sentence in the plural and using "they."

 Corrected Page 8

 I encourage you to continue to explore and expand on this exciting peer to peer intervention.

 Thank you for your kindness and valuable suggestion. 

 The purpose of this study is to examine the effect of exchanging paintings with peers on the happiness of children ages 7-11 years using a clinical trial. It sounds like an innovative technique to improve the happiness of children; however, the study design could not provide convincing results for health professionals to further investigate if this technique would really work. Therefore, better study design is needed to improve children’s psychological health.

Thanks for your great advice. The researchers noticed limitations during this study, too, which are mentioned in the Limitations section. Since this was the first experience of this kind of study in Iran, it certainly had some shortcomings that we hope will be eliminated in the future studies with improved designs. Nonetheless, we tried to clarify the ambiguities in the methods section during revision. 

Best: Sadat Hoseini

---

## [Decision Letter · Decision Letter 1]

13 Jan 2021

PONE-D-20-10349R1

The effect of exchanging drawings with peers on the happiness of children with cancer, ages 7-11 years: Clinical trial

PLOS ONE

Dear Dr. Sadat Hoseini,

Thank you for submitting your manuscript to PLOS ONE. After careful consideration, we feel that it has merit but does not fully meet PLOS ONE’s publication criteria as it currently stands. Therefore, we invite you to submit a revised version of the manuscript that addresses the points raised during the review process.

All the reviewers acknowledged the improvement in te paper, but still there are aspects (such the writing) that needs major work. Please consider are fully the suggestions of the reviewers.

We look forward to receiving your revised manuscript.

Kind regards,

Andrea Martinuzzi

Academic Editor

PLOS ONE

Reviewers' comments:

Reviewer's Responses to Questions

**Comments to the Author**

1. If the authors have adequately addressed your comments raised in a previous round of review and you feel that this manuscript is now acceptable for publication, you may indicate that here to bypass the “Comments to the Author” section, enter your conflict of interest statement in the “Confidential to Editor” section, and submit your "Accept" recommendation.

Reviewer #1: (No Response)

Reviewer #2: (No Response)

Reviewer #3: All comments have been addressed

2. Is the manuscript technically sound, and do the data support the conclusions?

Reviewer #1: Partly

Reviewer #2: Yes

Reviewer #3: Partly

3. Has the statistical analysis been performed appropriately and rigorously? 

Reviewer #1: No

Reviewer #2: Yes

Reviewer #3: Yes

4. Have the authors made all data underlying the findings in their manuscript fully available?

Reviewer #1: Yes

Reviewer #2: Yes

Reviewer #3: Yes

5. Is the manuscript presented in an intelligible fashion and written in standard English?

Reviewer #1: Yes

Reviewer #2: No

Reviewer #3: Yes

6. Review Comments to the Author

Reviewer #1: Comments

Table 1, if the expected cells less than 5 more than 20% e.g. 'variable 'interested in drawing'' chi-square test is not suitable. Decimal point for p value to be standardized. At least one decimal point for percentage value, Cohen D to be denoted in table footnote or in statistical analysis section for what comparison. 'No'' to be replaced with n.

Table 2, the row for the paired t-test to be placed in the last row after mean difference. Decimal point for p value, SD, t value to be standardized. (%95Confidence Interval to be written as 95% Confidence Interval. *T-Test to be replaced with *Independent t-test

Ensure all the statistical tests used in the study are stated in the statistical analysis section.

Reviewer #2: The revision brings more clarity regarding the intervention and other methods. Several instances of "painting" have not been changed to "drawing" e.g., page. 9, line 15, page 10, line 14, page 15, lines 10 and 12, and perhaps others. Gender bias remains, page 11, line 5. Also check for typos. In the discussion session, some examples are given e.g., reference 38) that were facilitated by art therapists, which could account for success because although it may look the same, the process and goals can be different. If art therapy examples are going to be used, the difference needs to be identified.

Reviewer #3: Thank you for addressing the reviewers comments in your paper. The revised version has improved a greatly deal comparing with the previous version. However, much improvement, especially in academic writing is needed for the future revision. I attached the detailed comments for your reference.

7. PLOS authors have the option to publish the peer review history of their article (what does this mean?). If published, this will include your full peer review and any attached files.

Reviewer #1: No

Reviewer #2: No

Reviewer #3: No

---

## [Author Response · Author response to Decision Letter 1]

18 Mar 2021

Response to reviewer

Dear Editor,

We are grateful for your meticulous review of this manuscript, and your suggestions, which have been very helpful in improving its quality. We also thank the reviewers for their careful reading and valuable suggestions. All the comments we received on this study have been addressed and replied separately. We hope that these changes to the manuscript will facilitate the decision to publish this study in your journal.

Reviewer 1

comment answer place

Table 1, if the expected cells less than 5 more than 20% e.g. 'variable 'interested in drawing'' chi-square test is not suitable. Decimal point for p value to be standardized. At least one decimal point for percentage value, Cohen D to be denoted in table footnote or in statistical analysis section for what comparison. 'No'' to be replaced with n. Was corrected Page 23 table 1

And file table

Table 2, the row for the paired t-test to be placed in the last row after mean difference. Decimal point for p value, SD, t value to be standardized. (%95Confidence Interval to be written as 95% Confidence Interval. *T-Test to be replaced with *Independent t-test Was corrected Page 23 table 2

And file table

Ensure all the statistical tests used in the study are stated in the statistical analysis section. Was added

using the independent t-test and paired t-test for comparison of the happiness scores between the two groups , the Chi-square test for comparison of the demographics, Fisher’s exact test for interest in drawing and paired t-test for comparison of the happiness scores in each group Page 13

Reviewer2

 The revision brings more clarity regarding the intervention and other methods. Several instances of "painting" have not been changed to "drawing" e.g., page. 9, line 15, page 10, line 14, page 15, lines 10 and 12, and perhaps others. All of them was changed

I hope did all of them and have not mistake whole of paper 

Gender bias remains, page 11, line 5. Also check for typos. Was corrected

The drawing intervention sessions were held after the child had received their medications and been visited by the doctors to avoid disturbing the child during the process of drawing.

 Page 11

In the discussion session, some examples are given e.g., reference 38) that were facilitated by art therapists, which could account for success because although it may look the same, the process and goals can be different. If art therapy examples are going to be used, the difference needs to be identified. Was corrected

Although the process and goals of drawing therapy and drawing are not similar, this research showed benefit of the process of drawing for reducing behavior problems. Page 16

Reviewer3

Thank you for addressing the reviewers comments in your paper. The revised version has improved a greatly deal comparing with the previous version. Thanks for attention 

However, much improvement, especially in academic writing is needed for the future revision. I attached the detailed comments for your reference. Was reedited 

Unfortunately I did not received attach file. I send some email but I cannot get it. Whole of paper

---

## [Decision Letter · Decision Letter 2]

31 Mar 2021

PONE-D-20-10349R2

The effect of exchanging drawings with peers on the happiness of children with cancer, aged 7-11 years: A clinical trial

PLOS ONE

Dear Dr. Sadat Hoseini,

Thank you for submitting your manuscript to PLOS ONE. After careful consideration, we feel that it has merit but does not fully meet PLOS ONE’s publication criteria as it currently stands. Therefore, we invite you to submit a revised version of the manuscript that addresses the points raised during the review process.

Please address the last minor changes requested by the reviewers.

Please submit your revised manuscript by April 15th If you will need more time than this to complete your revisions, please reply to this message or contact the journal office at plosone@plos.org. Please include the following items when submitting your revised manuscript:

We look forward to receiving your revised manuscript.

Kind regards,

Andrea Martinuzzi

Academic Editor

PLOS ONE

Journal Requirements:

Reviewers' comments:

Reviewer's Responses to Questions

**Comments to the Author**

1. If the authors have adequately addressed your comments raised in a previous round of review and you feel that this manuscript is now acceptable for publication, you may indicate that here to bypass the “Comments to the Author” section, enter your conflict of interest statement in the “Confidential to Editor” section, and submit your "Accept" recommendation.

Reviewer #1: All comments have been addressed

Reviewer #2: (No Response)

2. Is the manuscript technically sound, and do the data support the conclusions?

Reviewer #1: Partly

Reviewer #2: Yes

3. Has the statistical analysis been performed appropriately and rigorously? 

Reviewer #1: (No Response)

Reviewer #2: Yes

4. Have the authors made all data underlying the findings in their manuscript fully available?

Reviewer #1: No

Reviewer #2: Yes

5. Is the manuscript presented in an intelligible fashion and written in standard English?

Reviewer #1: Yes

Reviewer #2: Yes

6. Review Comments to the Author

Reviewer #1: Minor comments

The derived p value whether based on 1 or 2-tailed test to be stated.

List of references did not conform to the journal format.

Reviewer #2: Good revision! Instead of recommending "Minor Revision", my recommendation would more accurately be "Accept Pending." Just a few comments:

Page 12, line 5, “force” sounds too aggressive; suggest using “encourage.”

Page 14, line 8, change uppercase N to lower case n because it represents part of the total sample, N = 66, n =33.

Page 17, line 7-8, “brief the children’s teachers and parents on the study objectives…” Were the children told than an objective of the study was to make children with cancer happier?

Page 19, line 9, typo “ad” to “and”

7. PLOS authors have the option to publish the peer review history of their article (what does this mean?). If published, this will include your full peer review and any attached files.

Reviewer #1: No

Reviewer #2: No

---

## [Author Response · Author response to Decision Letter 2]

1 Apr 2021

Response to reviewer

Dear Editor,

We are grateful for your meticulous review of this manuscript, and your suggestions, which have been very helpful in improving its quality. We also thank the reviewers for their careful reading and valuable suggestions. All the comments we received on this study have been addressed and replied separately. We hope that these changes to the manuscript will facilitate the decision to publish this study in your journal.

Reviewer1

comments answer place

The derived p value whether based on 1 or 2-tailed test to be stated. Was corrected In statistical section and table 2 

List of references did not conform to the journal format. All references conform to the journal format and some mistake was corrected and one reference was deleted for duplication write References section 

Reviewer2 

Page 12, line 5, “force” sounds too aggressive; suggest using “encourage.” Was corrected Page 5 

Page 14, line 8, change uppercase N to lower case n because it represents part of the total sample, N = 66, n =33. Was corrected Page 7

Page 17, line 7-8, “brief the children’s teachers and parents on the study objectives…” Were the children told than an objective of the study was to make children with cancer happier? Base on the consent form of TUMS the research team must be explain purpose and subject of research to participant and they must be sign this form .” I, , have been explained about the purpose & objectives of this research and I am aware of them. The purpose& objectives are as follows: This Survey is about the Effect of Exchanging Paintings with Peers on the Happiness of Children with Cancer” The consent form was attached the time of submission.

If it is necessary, I can write on the manuscript this sentence.(in the box of answer). Currently, I write on the manuscript “. A meeting was then arranged to brief the children’s teachers and parents on the study objectives and methods”.

Page 19, line 9, typo “ad” to “and” Was corrected 

Journal Requirements:

Please review your reference list to ensure that it is complete and correct. If you have cited papers that have been retracted, please include the rationale for doing so in the manuscript text, or remove these references and replace them with relevant current references. Any changes to the reference list should be mentioned in the rebuttal letter that accompanies your revised manuscript. If you need to cite a retracted article, indicate the article’s retracted status in the References list and also include a citation and full reference for the retraction notice. All references was checked and conform to journal form. 

One of the references was deleted for retyping 

Some references were corrected and others were updated

All change show with highlighted References section

1. Jafroodi M. Epidemiologic evaluation of pediatric malignancies in 17 Shahrivar Hospital. Journal of Guilan University of Medecine Science. 2000;68:14-21. Persian.

2. Redig A, McAllister S. Breast cancer as a systemic disease: a view of metastasis: a view of metastasis. Journal of internal medicine. 2013;4(2):113-26.

3. Holder M, Coleman B, Singh K. Temperament and happiness in children in India. Journal of Happiness Studies. 2012;13(2):261-74.

4. Katz C, Hamama L. Draw me everything that happened to you”: Exploring children's drawings of sexual abuse. Children and Youth Services Review. 2013;35(3): 877-82.

5. Mehranfar M, Younesi J, Banihashem A. Effectiveness of mindfulness-based cognitive therapy on reduction of depression and anxiety symptoms in mothers of children with cancer. Iran J Cancer Prev. 2012;5(1):1-9. Epub 2012/01/01. PubMed PMID: 25780532; PubMed Central PMCID: PMCPMC4352519 Persian..

6. Verkooijen HM. [Happiness and cancer]. Ned Tijdschr Geneeskd. 2012;156(51):A5752. Epub 2012/12/20. PubMed PMID: 23249517.. Dutch.

7. Rogers MA, Zaragoza-Lao E. Happiness and Children’s Health: An Investigation of Art, Entertainment, and Recreation. Am J Public Health. 2003;93(2):288-9. doi: 10.2105/ajph.93.2.288.

8. Dickie K, Holder MD, Binfet JT. Happiness in Children. [cited 27 July 2017]. Availiable from: https://www.oxfordbibliographies.com/view/document/obo-9780199791231/obo-9780199791231-0188.xml.

9. Rollins J. Tell me about it: drawing as a communication tool for children with cancer. Journal of Pediatric Oncology Nursing. 2005;22(4): 203-21.

10. Badri M, Al Nuaimi A, Guang Y, Al Sheryani Y, Al Rashedi A. The effects of home and school on children’s happiness: a structural equation model. International Journal of Child Care and Education Policy. 2018;12(1):17. doi: 10.1186/s40723-018-0056-z.

11. Ackerman C. What is Positive Psychology & Why is It Important. [cited 06.12.2020]. Available from: https://positivepsychology.com/what-is-positive-psychology-definition/.

12. Bekhet AK, Zauszniewski JA, Nakhla WE. Happiness: theoretical and empirical considerations. Nurs Forum. 2008;43(1):12-23. Epub 2008/02/14. doi: 10.1111/j.1744-6198.2008.00091.x. PubMed PMID: 18269440.

13. Chaves C, Hervas G, García FE, Vazquez C. Building Life Satisfaction Through Well-Being Dimensions: A Longitudinal Study in Children with a Life-Threatening Illness. Journal of Happiness Studies. 2015;17(3):1051-67. doi: 10.1007/s10902-015-9631-y.

14. Hendriks T, Schotanus-Dijkstra M, Hassankhan A, de Jong J, Bohlmeijer E. The Efficacy of Multi-component Positive Psychology Interventions: A Systematic Review and Meta-analysis of Randomized Controlled Trials. Journal of Happiness Studies. 2019;21(1):357-90. doi: 10.1007/s10902-019-00082-1.

15. Kubzansky LD, Kim ES, Salinas J, Huffman JC, Kawachi I. Happiness, health, and mortality. Lancet. 2016;388(10039):27. Epub 2016/07/12. doi: 10.1016/S0140-6736(16)30896-0. PubMed PMID: 27397787.

16. Mahon NE, Yarcheski A. Alternative theories of happiness in early adolescents. Clin Nurs Res. 2002;11(3):306-23. Epub 2002/08/16. doi: 10.1177/10573802011003006. PubMed PMID: 12180642.

17. Rogers MA, Zaragoza-Lao E. Happiness and children's health: an investigation of art, entertainment, and recreation. Am J Public Health. 2003;93(2):288-9. Epub 2003/01/30. doi: 10.2105/ajph.93.2.288. PubMed PMID: 12554587; PubMed Central PMCID: PMCPMC1447731.

18. Veenhoven R. Healthy happiness: effects of happiness on physical health and the consequences for preventive health care. Journal of Happiness Studies. 2007;9(3):449-69. doi: 10.1007/s10902-006-9042-1.

19. Barfarazi H, Pourghaznein T, Mohajer S, Mazlom SR, Asgharinekah SM. Evaluating the Effect of Painting Therapy on Happiness in the Elderly. Evidence Based Care. 2018;8(3):17-26.

20. Askary P, Khayat A. The effectiveness of music therapy on severity of pain, perceived stress and happiness in adolescents with leukemia. . Positive Psychology. 2018;3(4):15-27.

21. Mirzaie M, Navidi Z. Survey personal and disease characteristics of children with cancer hospitalized in 17 shahrivar hospital. Journal of Guilan Faculty of Medicine. 2010;19(61):32-6.

22. Oliveira Pinto SM, Alves Caldeira Berenguer SM, Martins JC. Cancer, Health Literacy, and Happiness: Perspectives from Patients under Chemotherapy. Nurs Res Pract. 2013;2013:291767. Epub 2013/10/04. doi: 10.1155/2013/291767. PubMed PMID: 24089635; PubMed Central PMCID: PMCPMC3780657.

23. Khodabakhshi Koolaee A, Vazifehdar R, F, B., Akbari M. Impact of painting therapy on aggression and anxiety of children with cancer. Caspian J Pediatr. 2016;2(2):135-41.

24. Zareapour A, Falahi Khoshknab M, Kashaninia Z, Biglarian A. Effect of group play therapy on depression in children with cancer. Scientific Journal of Kurdistan University of Medical Sciences,. 2009;14(3):64-72. Persian.

25. Borzou R, Bayat Z, Salvati M, Homayounfar S. A comparison of Individual and Peer Educational Methods on Quality of life in patients with heart failure. Iranian Journal of Medical Education. 2014;14(9):767-76. Persian.

26. Philis-Tsimikas A, Fortmann A, Lleva-Ocana L, Walker C, Gallo LC. Peer-led diabetes education programs in high-risk Mexican Americans improve glycemic control compared with standard approaches: a Project Dulce promotora randomized trial. Diabetes Care. 2011;34(9):1926-31. Epub 2011/07/22. doi: 10.2337/dc10-2081. PubMed PMID: 21775748; PubMed Central PMCID: PMCPMC3161298.

27. Motevaselian M, Nasiriani K. Impact of Near-peer teaching on Learning Dressing Skill among Nursing Students. Iranian Journal of Medical Education. 2014;14(8):670-6. Persian.

28. Heisler M. Overview of peer support models to improve diabetes self-management and clinical outcomes. Diabetes Spectrum. 2007;20(4):214-21.

29. Khaw C, Raw L. The outcomes and acceptability of near-peer teaching among medical students in clinical skills. International Journal of Medical Education. 2016;7:188-94.

30. Priharjo R, Hoy G. Use of peer teaching to enhance student and patient education. Nursing Standard. 2011;25(20):3-40.

31. Michalos AC, editor. Encyclopedia of Quality of Life and Well-Being Research. 1nd ed. Dordrecht: Springer Netherlands; 2014. p. 6622-5.

32. Durualp E, Altay N. A Comparison of Emotional Indicators and Depressive Symptom Levels of School-Age Children With and Without Cancer. Journal of Pediatric Oncology Nursing. 2012;29(4):232-9.

33. Hockenberry M, Wilson D, Wong D. Wong's Essentials of Pediatric Nursing. 10th ed. U.S: Mosby; 2016.

34. Michel G, Rebholz CE, von der Weid NX, Bergstraesser E, Kuehni CE. Psychological distress in adult survivors of childhood cancer: the Swiss Childhood Cancer Survivor study. J Clin Oncol. 2010;28(10):1740-8. Epub 2010/03/03. doi: 10.1200/JCO.2009.23.4534. PubMed PMID: 20194864.

35. Sanjari M, Jafarppour M, Safarabadi T, Hosseini F. Coping with cancer in teenagers and their parents. Iran Journal of Nursing. 2005;18(41):111-22. Persian.

36. Beebe A, Gelfand EW, Bender B. A randomized trial to test the effectiveness of art therapy for children with asthma. J Allergy Clin Immunol. 2010;126(2):263-6, 6 e1. Epub 2010/05/14. doi: 10.1016/j.jaci.2010.03.019. PubMed PMID: 20462632.

37. Lee S-L, Liu H-LA. A pilot study of art therapy for children with special educational needs in Hong Kong. The Arts in Psychotherapy. 2016;51(51):24-9. doi: 10.1016/j.aip.2016.08.005.

38. Barrera ME, Rykov MH, Doyle SL. The effects of interactive music therapy on hospitalized children with cancer: a pilot study. Psychooncology. 2002;11(5):379-88. Epub 2002/09/14. doi: 10.1002/pon.589. PubMed PMID: 12228871.

39. Gariépy N, Howe N. The therapeutic power of play: examining the play of young children with leukaemia. Child: care, health and development. 2003;29(6):523-37.

---

## [Editor Report · Decision Letter 3]

14 Sep 2021

The effect of exchanging drawings with peers on the happiness of children with cancer, aged 7-11 years: A clinical trial

PONE-D-20-10349R3

Dear Dr. Sadat Hoseini,

We’re pleased to inform you that your manuscript has been judged scientifically suitable for publication and will be formally accepted for publication once it meets all outstanding technical requirements.

Kind regards,

Andrea Martinuzzi

Academic Editor

PLOS ONE
---

## [Editor Report · Acceptance letter]

7 Oct 2021

PONE-D-20-10349R3 

The effect of exchanging drawings with peers on the happiness of children with cancer, aged 7-11 years: A clinical trial 

Dear Dr. Sadat Hoseini:

I'm pleased to inform you that your manuscript has been deemed suitable for publication in PLOS ONE. Congratulations! Your manuscript is now with our production department. 

Kind regards, 

on behalf of

Dr. Andrea Martinuzzi 

Academic Editor

PLOS ONE